# Deepcode: Feedback Codes via Deep Learning

**Hyeji Kim**[*], **Yihan Jiang**[†], **Sreeram Kannan**[†], **Sewoong Oh**[‡], **Pramod Viswanath**[‡]

Samsung AI Centre Cambridge[*], University of Washington[†], University of Illinois at Urbana Champaign[‡]

## Abstract

The design of codes for communicating reliably over a statistically well defined channel is an important endeavor involving deep mathematical research and wide-ranging practical applications. In this work, we present the first family of codes obtained via deep learning, which significantly beats state-of-the-art codes designed over several decades of research. The communication channel under consideration is the Gaussian noise channel with feedback, whose study was initiated by Shannon; feedback is known theoretically to improve reliability of communication, but no practical codes that do so have ever been successfully constructed.

We break this logjam by integrating information theoretic insights harmoniously with recurrent-neural-network based encoders and decoders to create novel codes that outperform known codes by 3 orders of magnitude in reliability. We also demonstrate several desirable properties in the codes: (a) generalization to larger block lengths; (b) composability with known codes; (c) adaptation to practical constraints. This result also presents broader ramifications to coding theory: even when the channel has a clear mathematical model, deep learning methodologies, when combined with channel-specific information-theoretic insights, can potentially beat state-of-the-art codes, constructed over decades of mathematical research.

## 1 Introduction

The ubiquitous digital communication enabled via wireless (e.g. WiFi, mobile, satellite) and wired (e.g. ethernet, storage media, computer buses) media has been the plumbing underlying the current information age. The advances of reliable and efficient digital communication have been primarily driven by the design of codes which allow the receiver to recover messages reliably and efficiently under noisy environments. The discipline of coding theory has made significant progresses in the past seven decades since Shannon's celebrated work in 1948 [1]. As a result, we now have near optimal codes in a canonical setting, namely, Additive White Gaussian Noise (AWGN) channel. However, several channel models of great practical interest lack efficient and practical coding schemes.

Channels with *feedback* (from the receiver to the transmitter) is an example of a long-standing open problem and with significant practical importance. Modern wireless communication includes feedback, in one form or the other; for example, the feedback can be the received value itself or quantization of the received value or an automatic repeat request (ARQ) [2]. Accordingly, there are different models for channels with feedback, and among them, the AWGN channel with *output* feedback is a model that captures the essence of channels with feedback; this model is also classical, introduced by Shannon in 1956 [3]. In this channel model, the received value is fed back (with

---

[*]H. Kim is with Samsung AI Centre Cambridge, UK. Email: `hkim1505@gmail.com`

[†]Y. Jiang and S. Kannan are with the Department of Electrical Engineering at University of Washington. Email: `yihanrogerjiang@gmail.com` and `ksreeram@uw.edu`.

[‡]S. Oh and P. Viswanath are with the Coordinated Science Lab at University of Illinois at Urbana Champaign (UIUC). S. Oh is with the Department of Industrial and Enterprise Systems Engineering at UIUC. P. Viswanath is with the Department of Electrical Engineering at UIUC. Email: `{swoh,pramodv}@illinois.edu`

unit time delay) to the transmitter without any processing (we refer to Figure 1 for an illustration of channel). Designing codes for this channel via deep learning approaches is the central focus of this paper.

While the output feedback does not improve the Shannon capacity of the AWGN channel [3], it is known to provide better reliability at finite block lengths [4]. On the other hand, practical coding schemes have not been successful in harnessing the feedback gain thereby significantly limiting the use of feedback in practice. This state of the art is at odds with the theoretical predictions of the gains in reliability via using feedback: the seminal work of Schalkwijk-Kailath [4] proposed S-K scheme, a (theoretically) achievable scheme with superior reliability guarantees, but which suffers from extreme sensitivity to both the precision of the numerical computation and noise in the feedback [5, 6]. Another competing scheme of [7] is designed for channels with noisy feedback, but not only is the reliability poor, it is almost independent of the feedback quality, suggesting that the feedback data is not being fully exploited. More generally, it has been proven that no *linear* code incorporating the noisy output feedback can perform well [8]. This is especially troubling since all practical codes are linear and linear codes are known to achieve capacity (without feedback) [9].

In this paper, we demonstrate new neural network-driven encoders (with matching decoders) that operate significantly better (100–1000 times) than state of the art, on the AWGN channel with (noisy) output feedback. We show that architectural insights from simple communication channels with feedback when coupled with recurrent neural network architectures can discover novel codes. We consider Recurrent Neural Network (RNN) parameterized encoders (and decoders), which are inherently *nonlinear* and map information bits *directly* to real-valued transmissions in a sequential manner.

Designing codes driven by deep learning has been of significant interest recently [10–25], starting from [10] which proposes an autoencoder framework for communications. In [10], it is demonstrated that for classical AWGN channels, feedforward neural codes can mimic the performance of a well-known code for a short block length (4 information bits). Extending this idea to orthogonal frequency division multiplex (OFDM), [11, 12] show that neural codes can mimic the performance of state-of-the-art codes for short block lengths (8 information bits). Several results extend the autoencoder idea to other settings of AWGN channels [13] and modulation [26]). Beyond AWGN channels, [14] considers the problem of communicating a complicated source (text) over erasure channels and shows that RNN-based neural codes that map raw texts directly to a codeword can beat the state-of-the art codes, when the reliability is evaluated by human (as opposed to bit error rate). Deep learning has been applied also in the problem of designing decoders for existing encoders [15–19], demonstrating the efficiency, robustness, and adaptivity of neural decoders over the existing decoders. In a different context, for distributed computation, where encoder adds redundant computations so that the decoder can reliably approximate the desired computations under unavailabilities, [20] showed that neural network based codes can beat the state of the art codes.

While several works in the past years apply deep learning for channel coding, very few of them consider the design of novel codes using deep learning (rather than decoders). Furthermore, none of them are able to beat state-of-the-art codes on a standard (well studied) channel. We demonstrate first family of codes obtained via deep learning which beats state-of-the-art codes, signaling a potential shift in code design, which historically has been driven by individual human ingenuity with sporadic progress over the decades. Henceforth, we call this new family of codes *Deepcode*. We also demonstrate the superior performance of variants of Deepcode under a variety of practical constraints. Furthermore, Deepcode has complexity comparable to traditional codes, even without any effort at optimizing the storage and run-time complexity of the neural network architectures. Our main contributions are as follows:

1. We demonstrate Deepcode – a new family of RNN-driven neural codes that have *three orders of magnitude* better reliability than state of the art with both noiseless and noisy feedback. Our results are significantly driven by the intuition obtained from information and coding theory, in designing a series of progressive improvements in the neural network architectures (Section 3 and 4).

2. We show that variants of Deepcode significantly outperform state-of-the art codes under a variety of practical constraints (example: delayed feedback, very noisy feedback link) (Section 4).

3. We show *composability*: Deepcode naturally concatenates with a traditional inner code and demonstrates continued improvements in reliability as the block length increases (Section 4).

4. Our interpretation and analysis of Deepcode provide guidance on the fundamantal understanding of how the feedback can be used and some information theoretic insights into designing codes for channels with feedback (Section 5).

## 2  Problem formulation

The most canonical channel studied in the literature (example: textbook material [27]) and also used in modeling practical scenarios (example: 5G LTE standards) is the Additive White Gaussian Noise (AWGN) channel *without* feedback. Concretely, the encoder takes in $K$ information bits jointly, $\mathbf{b} = (b_1, \cdots, b_K) \in \{0,1\}^K$, and outputs $n$ real valued signals to be transmitted over a noisy channel (sequentially). At the $i$-th transmission for each $i \in \{1, \ldots, n\}$, a transmitted symbol $x_i \in \mathbb{R}$ is corrupted by an independent Gaussian noise $n_i \sim \mathcal{N}(0, \sigma^2)$, and the decoder receives $y_i = x_i + n_i \in \mathbb{R}$. After receiving the $n$ received symbols, the decoder makes a decision on which information bit $\mathbf{b}$ was sent, out of $2^K$ possible choices. The goal is to maximize the probability of correctly decoding the received symbols and recover $\mathbf{b}$.

Both the encoder and the decoder are functions, mapping $\mathbf{b} \in \{0,1\}^K$ to $\mathbf{x} \in \mathbb{R}^n$ and $\mathbf{y} \in \mathbb{R}^n$ to $\hat{\mathbf{b}} \in \{0,1\}^K$, respectively. The design of a good code (an encoder and a corresponding decoder) addresses both $(i)$ the statistical challenge of achieving a small error rate; and $(ii)$ the computational challenge of achieving the desired error rate with efficient encoder and decoder. Almost a century of progress in this domain of coding theory has produced several innovative codes that efficiently achieve small error rate, including convolutional codes, Turbo codes, LDPC codes, and polar codes. These codes are known to perform close to the fundamental limits on reliable communication [28].

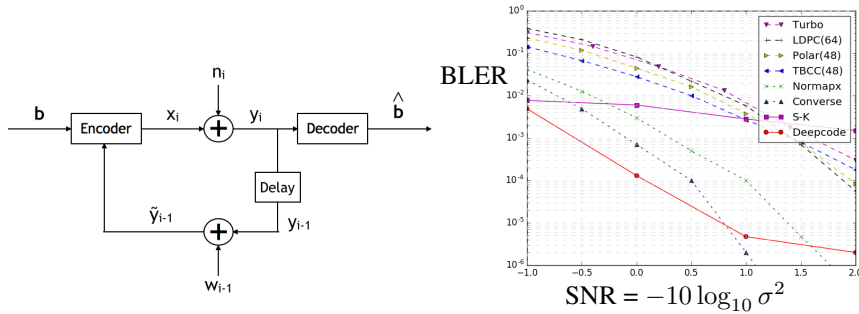

Figure 1: AWGN channel with noisy feedback (left). Deepcode significantly outperforms the baseline of S-K and the state-of-the art codes, on block-length 50 and noiseless feedback (right).

In a canonical AWGN channel *with* noisy feedback, the received symbol $y_i$ is transmitted back to the encoder after one unit time of delay and via another additive white Gaussian noise *feedback channel* (Figure 1). The encoder can use this feedback symbol to sequentially and adaptively decide what symbol to transmit next. At time $i$ the encoder receives a noisy view of what was received at the receiver (in the past by one unit time), $\tilde{y}_{i-1} = y_{i-1} + w_{i-1} \in \mathbb{R}$, where the noise is independent and distributed as $w_{i-1} \sim \mathcal{N}(0, \sigma_F^2)$. Formally, an *encoder* is now a function that sequentially maps the information bit vector $\mathbf{b}$ and the feedback symbols $\tilde{y}_1^{i-1} = (\tilde{y}_1, \cdots, \tilde{y}_{i-1})$ received thus far to a transmit symbol $x_i$: $f_i : (\mathbf{b}, \tilde{y}_1^{i-1}) \mapsto x_i, \quad i \in \{1, \cdots, n\}$ and a *decoder* is a function that maps the received sequence $y_1^n = (y_1, \cdots, y_n)$ into estimated information bits: $g : y_1^n \mapsto \hat{\mathbf{b}} \in \{0,1\}^K$.

The standard measures of performance are the average bit error rate (BER) defined as $\text{BER} \equiv (1/K) \sum_{i=1}^K \mathbb{P}(b_i \neq \hat{b}_i)$ and the block error rate (BLER) defined as $\text{BLER} \equiv \mathbb{P}(\mathbf{b} \neq \hat{\mathbf{b}})$, where the randomness comes from the forward and feedback channels and any other sources of randomness that might be used in the encoding and decoding processes. It is standard (both theoretically and practically) to have an average power constraint, i.e., $(1/n)\mathbb{E}[\|\mathbf{x}\|^2] \leq 1$, where $\mathbf{x} = (x_1, \cdots, x_n)$ and the expectation is over the randomness in choosing the information bits $\mathbf{b}$ uniformly at random, the randomness in the noisy feedback symbols $\tilde{y}_i$'s, and any other randomness used in the encoder.

While the capacity of the channel remains the same in the presence of feedback [3], the reliability can increase significantly as demonstrated by the celebrated result of Schalkwijk and Kailath (S-K), [4], which is described in detail in Appendix D. Although the optimal theoretical performance is met by the S-K code, critical drawbacks make it fragile. Theoretically, the scheme critically relies on exactly noiseless feedback (i.e. $\sigma_F^2 = 0$), and does not extend to channels with even arbitrarily small amount of noise in the feedback (i.e. $\sigma_F^2 > 0$). Practically, the scheme is extremely sensitive to numerical precisions; we see this in Figure 1, the numerical errors dominate the performance of the S-K scheme, with a practical choice of MATLAB implementation with a precision of 16 bits to represent floating-point numbers.

Even with a noiseless feedback channel with $\sigma_F^2 = 0$, which the S-K scheme is designed for, it is outperformed significantly by our proposed Deepcode (described in detail in Section 3). At moderate SNR of 2 dB, Deepcode can outperform S-K scheme by three orders of magnitude in BLER. The resulting BLER is shown as a function of the Signal-to-Noise Ratio (SNR) defined as $-10\log_{10}\sigma^2$. Also shown as baselines are the state-of-the art polar, LDPC, and tail-bitting convolutional codes (TBCC) in a 3GPP document for the 5G meeting [29] (we refer to Appendix A for the details of these codes used in the simulation). Deepcode significantly improves over all state-of-the-art codes of similar block-length and the same rate. Also plotted as a baseline are the theoretically estimated performance of the best code with no efficient decoding schemes. This impractical baseline lies between approximate achievable BLER (labelled Normapx in the figure) and a converse to the BLER (labelled Converse in the figure) from [28, 30]. More recently proposed schemes address S-K scheme's sensitivity to noise in the feedback, but either still suffer from similar sensitivity to numerical precisions at the decoder [31], or is incapable of exploiting the feedback information [7] as we illustrate in Figure 4.

## 3 Neural encoder and decoder

A natural strategy to create a feedback code is to utilize a recurrent neural network (RNN) as an encoder since $(i)$ communication with feedback is naturally a sequential process and $(ii)$ to exploit the sequential structure for efficient decoding. We propose representing the encoder and the decoder as RNNs, training them jointly under AWGN channels with noisy feedback, and minimizing the error in decoding the information bits. However, in our experiments, we find that this strategy by itself is insufficient to achieve any performance improvement with feedback.

We exploit information theoretic insights to enable improved performance, by considering the erasure channel with feedback: here transmitted bits are either received perfectly or erased and whether the previous bit was erased or received perfectly is fed back to the transmitter. In such a channel, the following two-phase scheme can be used: transmit a block of symbols, and then transmit whichever symbols were erased in the first block (and ad infinitum). This motivates a two-phase scheme, where uncoded bits are sent in the first phase, and then based on the feedback in the first phase, coded bits are sent in the second phase; thus the code only needs to be designed for the second phase. Even inside this two-phase paradigm, several architectural choices need to be made. We show in this section that these intuitions can be critically employed to innovate neural network architectures.

Our experiments focus on the setting of rate 1/3 and information block length of 50 for concreteness[1]. That is, the encoder maps $K = 50$ message bits to a codeword of length $n = 150$. We discuss generalizations to longer block lengths in Section 4.

**A. RNN feedback encoder/decoder (RNN (linear) and RNN (tanh)).** We propose an encoding scheme that progresses in two phases. In the first phase, the $K$ information bits are sent raw (uncoded) over the AWGN channel. In the second phase, $2K$ coded bits are generated based on the information bits **b** and (delayed) output feedback and sequentially transmitted. We propose a decoding scheme using two layers of bidirectional Gated Recurrent Units (GRU). When jointly trained, a *linear* RNN encoder achieves performance close to Turbo code that does not use the feedback information at all as shown in Figure 2. With a *non-linear* activation function of $\tanh(\cdot)$, the performance improves, achieving BER close to the existing S-K scheme. Such a gain of non-linear codes over linear ones is in-line with theory [31].

Encoder A: RNN feedback encoder

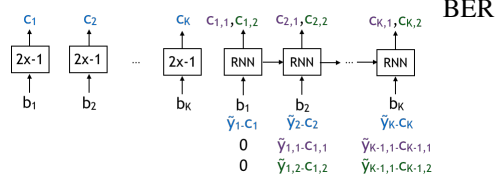

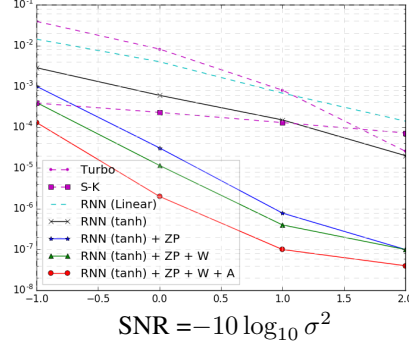

BER

$\mathrm{SNR} = -10 \log_{10} \sigma^2$

Figure 2: Building upon a simple linear RNN encoder (left), we progressively improve the architecture. Eventually with RNN(tanh)+ZP+W+A architecture formally described in Section 3, we significantly outperform the baseline of S-K scheme and Turbo code, by several orders of magnitude in the bit error rate, on block-length 50 and noiseless feedback ($\sigma_F^2 = 0$).

**Encoding.** The architecture of the encoder is shown in Figure 2. The encoding process has two phases. In the first phase, the encoder simply transmits the $K$ raw message bits. That is, the encoder maps $b_k$ to $c_k = 2b_k - 1$ for $k \in \{1, \cdots, K\}$, and stores the feedback $\tilde{y}_1, \cdots, \tilde{y}_K$ for later use. In the second phase, the encoder generates a coded sequence of length $2K$ (length $(1/r - 1)K$ for general rate $r$ code) through a single directional RNN. In particular, each $k$-th RNN cell generates two coded bits $c_{k,1}, c_{k,2}$ for $k \in \{1, \ldots, K\}$, which uses both the information bits and (delayed) output feedback from the earlier raw information bit transmissions. The input to the $k$-th RNN cell is of size four: $b_k$, $\tilde{y}_k - c_k$ (the estimated noise added to the $k$-th message bit in phase 1) and the most recent two noisy feedbacks from phase 2: $\tilde{y}_{k-1,1} - c_{k-1,1}$ and $\tilde{y}_{k-1,2} - c_{k-1,2}$. Note that we use $\tilde{y}_{k,j} = c_{k,j} + n_{k,j} + w_{k,j}$ to denote the feedback received from the transmission of $c_{k,j}$ for $k \in \{1, \cdots, K\}$ and $j \in \{1, 2\}$, and $n_{k,j}$ and $w_{k,j}$ are corresponding forward and feedback channel noises, respectively.

To generate codewords that satisfy power constraint, we put a normalization layer to the RNN outputs so that each coded bit has a mean 0 and a variance 1. During training, the normalization layer subtracts the batch mean from the output of RNN and divide by the standard deviation of the batch. After training, we compute the mean and the variance of the RNN outputs over $10^6$ examples. In testing, we use the precomputed means and variances. Further implementation details are in Appendix B.

**Decoding.** Based on the received sequence $\mathbf{y} = (y_1, \cdots, y_k, y_{1,1}, y_{1,2}, y_{2,1}, y_{2,2}, \cdots, y_{K,1}, y_{K,2})$ of length $3K$, the decoder estimates $K$ information bits. For the decoder, we use a two-layered bidirectional Gated Recurrent Unit (GRU), where the input to the $k$-th GRU cell is a tuple of three received symbols, $(y_k, y_{k,1}, y_{k,2})$. We refer to Appendix B for more implementation details.

**Training.** Both the encoder and decoder are trained *jointly* using binary cross-entropy as the loss function over $4 \times 10^6$ examples, with batch size 200, via an Adam optimizer ($\beta_1$=0.9, $\beta_2$=0.999, $\epsilon$=1e-8). The input to the neural network is $K$ information bits and the output is $K$ estimated bits (as in the autoencoder setting). During the training, we let $K = 100$. AWGN channels are simulated for the channels from the encoder to the decoder and from decoder to the encoder. In training, we let the forward SNR equal to be test SNR and feedback SNR to be the test feedback SNR. We randomly initialize weights of the encoder and the decoder. We observed that training with random initialization of encoder-decoder gives a better encoder-decoder compared to initializing with a pre-trained encoder/decoder by sequential channel codes for non-feedback AWGN channels (e.g. convolutional codes). We also use a decaying learning rate and gradient clipping; we reduce the learning rate by 10 times after training with $10^6$ examples, starting from 0.02. Gradients are clipped to 1 if $L_2$ norm of the gradient exceeds 1 so that we prevent the gradients from getting too large.

*Typical error analysis.* Due to the recurrent structure in generating coded bits $(c_{k,1}, c_{k,2})$, the coded bit stream carries more information on the first few bits than last few bits (e.g. $b_1$ than $b_K$). This results in more errors in the last information bits, as shown in Figure 3, where we plot the average BER of $b_k$ for $k = \{1, \cdots, K\}$.

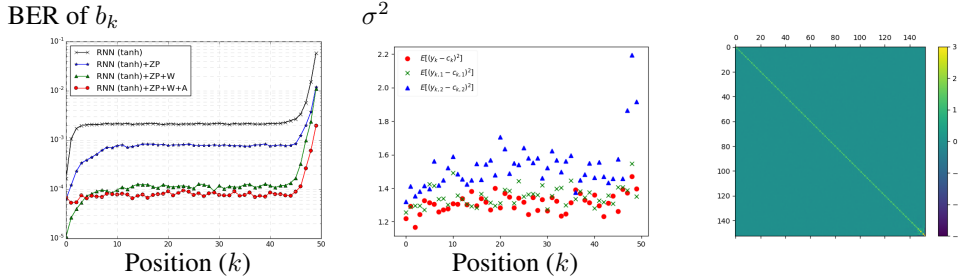

Figure 3: (Left) A naive RNN(tanh) code gives a high BER in the last few information bits. With the idea of zero padding and power allocation, the RNN(tanh)+ZP+W+A architecture gives a BER that varies less across the bit position, and overall BER is significantly improved over the naive RNN(tanh) code. (Middle) Noise variances across bit position which results in a block error: High noise variance on the second parity bit stream $(c_{1,2}, \cdots, c_{K,2})$ causes a block error. (Right) Noise covariance: Noise sequence which results in a block error does not have a significant correlation across position.

**B. RNN feedback code with zero padding (RNN (tanh) + ZP).** In order to reduce high errors in the last information bits, as shown in Figure 3, we apply the zero padding (ZP) technique; we pad a zero in the end of information bits, and transmit a codeword for the padded information bits (The encoder and decoder with zero padding are illustrated in Appendix B). By bringing zero padding, the BER of the last information bits, as well as other bits, drops significantly, as shown in Figure 3. Zero padding requires a few extra channel usages (e.g. with 3 symbol zero padding, we map 50 information bits to a codeword of length 153). However, due to the significant improvement in BER, it is widely used in sequential codes (e.g. convolutional codes and turbo codes).

*Typical error analysis.* To see if there is a pattern in the noise sequence which makes the decoder fail, we study the first and second order noise statistics which result in the error in decoding. In Figure 3 (Middle), we plot the average variance of noise added to $b_k$ in first phase and $c_{k,1}$ and $c_{k,2}$ in the second phase, as a function of $k$. From the figure, we make two observations; $(i)$ large noise in the last bits cause an error, and $(ii)$ large noise in $c_{k,2}$ is likely to cause an error, which implies that the raw bit stream and the coded bit streams are not equally robust to the noise – an observation that will be exploited next. In Figure 3 (Right), we plot noise covariances that result in a decoding error. From Figure 3 (Right), we see that there is no particular correlation within the noise sequence that makes the decoder fail. This suggests that there is no particular error pattern to be exploited, and the BER performance further improved.

**C. RNN feedback code with power allocation (RNN(tanh) + ZP + W).** Based on the observation that the raw bit $c_k$ and coded bit $c_{k,1}, c_{k,2}$ are not equally robust, as shown in Figure 3 (Middle), we introduce trainable weights which allow allocating different amount of power to the raw bit and coded bits. Appendix B provides all implementation details. By introducing and training these weights, we achieve the improvement in BER as shown in Figures 2 and 3.

*Typical error analysis.* While the average BER is improved by about an order of magnitude for most bit positions as shown in Figure 3 (Left), the BER of the last bit remains about the same. On the other hand, the BER of first few bits are now smaller, suggesting the following bit-specific power allocation method.

**D. Deepcode: RNN feedback code with bit power allocation (RNN(tanh) + ZP + W + A).** We introduce a weight vector allowing the power of bits in different position to be different, as illustrated in Figure 10. Ideally, we would like to reduce the power for the first information bits and increase the power for the last information bits. The resulting BER curve is shown in Figure 2(-o-). We can see that the BER is noticeably decreased. In Figure 3(-o-), we can see that the BER in the last bits are reduced, and we can also see that the BER in the first bits are increased. Our use of unequal power allocation across information bits is in-line with other approaches from information theory [32], [33]. We call this neural code Deepcode.

*Typical error analysis.* As shown in Figure 3, the BER at each position remains about the same except for the last few bits. This suggests a symmetry in our code and nearest-neighbor-like decoder. For an AWGN channel without feedback, it is known that the optimal decoder (nearest neighbor

decoder) under a symmetric code (in particular, each coded bit follows a Gaussian distribution) is robust to the distribution of noise [34]; the BER does not increase if we keep the power of noise and only change the distribution. As an experiment demonstrating the robustness of Deepcode, in Appendix E, we show that BER of Deepcode does not increase if we keep the power of noise and change the distribution from i.i.d. Gaussian to bursty Gaussian noise.

**Complexity.** Complexity and latency, as well as reliability, are important metrics in practice, as the encoder and decoder need to run in real time on mobile devices. Deepcode has computational complexity and latency comparable to currently used codes (without feedback) that are already in communication standards. Turbo decoder, for example, is a belief-propagation decoder with many (e.g., $10 - 20$) iterations, and each iteration is followed by a permutation. Turbo encoder also includes a permutation of information bits (of length $K$). On the other hand, the proposed neural encoder in Deepcode is a single layered RNN encoder with 50 hidden units, and the neural decoder in Deepcode is a 2-layered GRU decoder, also with 50 hidden units, all of which are matrix multiplications that can be parallelized. Ideas such as knowledge distillation [35] and network binarization [36] can be used to potentially further reduce the complexity of the network.

# 4 Practical considerations: noise, delay, coding in feedback, and blocklength

We considered so far the AWGN channel with noiseless output feedback with a unit time-step delay. In this section, we demonstrate the robustness of Deepcode (and its variants) under two variations on the feedback channel, *noise* and *delay*, and present generalization to longer block lengths. We show that ($a$) Deepcode and its variant that allows a $K$-step delayed feedback are more reliable than the state-of-the-art schemes in channels with *noisy* feedback; ($b$) by allowing the receiver to feed back an RNN encoded output instead of its raw *output*, and learning this RNN encoder, we achieve a further improvement in reliability, demonstrating the power of encoding in the feedback link; ($c$) Deepcode concatenated with turbo code achieves superior error rate decay as block length increases with noisy feedback.

**Noisy feedback.** We show that Deepcode trained under AWGN channels with *noisy* output feedback, achieves a significantly smaller BER than both S-K and C-L schemes under AWGN channels with *noisy* output feedback. In Figure 4 (Left), we plot the BER as a function of the feedback SNR for S-K scheme, C-L scheme, and Deepcode for a rate 1/3, 50 information bits, where we fix the forward channel SNR to be 0dB. As feedback SNR increases, we expect the BER to decrease. However, as shown in Figure 4 (Left), the C-L scheme, which is designed for noisy feedback, and S-K scheme are very sensitive to noise in the feedback, and reliability is almost independent of feedback quality. Deepcode outperforms these two baseline (linear) codes by a large margin, with decaying error as feedback SNR increases, showing that Deepcode harnesses *noisy* feedback information to make communication more reliable. This is highly promising as the performance under noisy feedback is directly related to the practical communication channels.

**Noise feedback with delay.** We model the practical constraint of *delay* in the feedback, by introducing a variant of Deepcode that works with a $K$ time-step delayed feedback (discussed in detail in Appendix B.5); recall $K$ is the number of information bits and this code tolerates a large delay in the feedback. Perhaps unexpectedly, we see from Figure 4 (Left) that this neural code for delayed feedback achieves a similar BER to no delay in the feedback; this is true especially at small feedback SNRs, till around 12dB.

**Noisy feedback with delay and coding.** It is natural to allow the receiver send back a general *function* of its past received values, i.e., receiver encodes its output and sends the coded (real valued) bit. Designing the code for this setting is challenging as it involves designing two encoders and one decoder jointly in a sequential manner. We propose using RNN as an encoder that maps noisy output to the transmitted feedback, with implementation details in Appendix B.5. Figure 4 demonstrates the powerful encoding of the received output, as learnt by the neural architecture; the BER is improved two-three times.

**Generalization to longer block lengths.** In wireless communications, a wide range of blocklengths are of interest (e.g., 40 to 6144 information bits in LTE standards). In previous sections, we considered block length of $50$ information bits. Here we show how to generalize Deepcode to longer block lengths and achieve an improved reliability as we increase the block length.

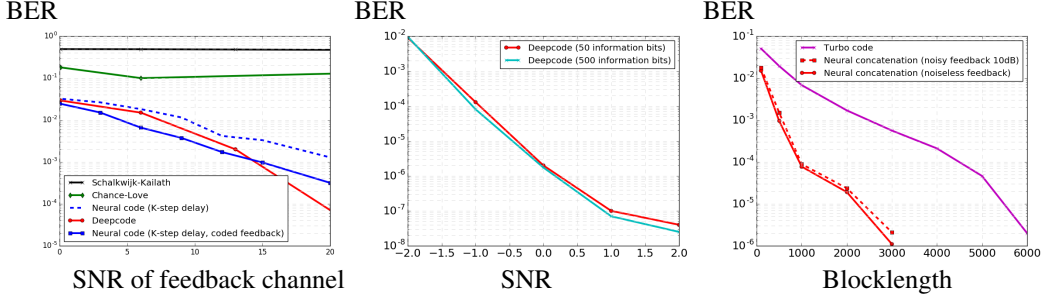

BER — SNR of feedback channel | BER — SNR | BER — Blocklength

Figure 4: (Left) Deepcode (introduced in section 3) and a variant of neural code which allows $K$ time-step delay significantly outperform the two baseline schemes under noisy feedback. Another variant of Deepcode which allows the receiver to feed back an *RNN encoded* output (with $K$-step delay) performs even better than the neural code with *raw* output feedback (with unit-delay), demonstrating the power of coding in the feedback. (Middle) By unrolling the RNN cells of Deepcode, the BER of Deepcode remains the same for block lengths 50 to 500. (Right) Concatenated Deepcode and turbo code (with and without noise in the feedback) achieves BER that decays exponentially as block length increases, faster than turbo codes (without feedback) at same rate.

A natural generalization of the RNN-based neural code is to unroll the RNN cells. In Figure 4 (Middle), we plot the BER as a function of the SNR, for 50 information bits and length 500 information bits (under noiseless feedback) when we unroll the RNN cells. We can see that the BER remains the same as we increase block lengths. This is not an entirely satisfying generalization because, typically, it is possible to design a code for which error rate decays faster as block length increases. For example, turbo codes have error rate decaying exponentially ($\log$ BER decades linearly) in the block length as shown in Figure 4 (Right). This critically relies on the interleaver, which creates long range dependencies between information bits that are far apart in the block. Given that the neural encoder is a sequential code, there is no strong long range dependence. Each transmitted bit depends on only a few past information bits and their feedback (we refer to Section 5 for a detailed discussion).

To resolve this problem, we propose a new concatenated code which concatenates Deepcode (as inner code) and turbo code as an outer code. The outer code is not restricted to a turbo code, and we refer to Appendix C for a detailed discussion.

In Figure 4 (Right), we plot the BERs of the concatenated code, under both noiseless and noisy feedback (of feedback SNR 10dB), and turbo code, both at rate $1/9$ at (forward) SNR $-6.5$dB. From the figure, we see that even with noisy feedback, BER drops almost exponentially ($\log$ BER drops linearly) as block length increases, and the slope is sharper than the one for turbo codes. We also note that in this setting, C-L scheme suggests not using the feedback.

## 5 Interpretation

Thus far we have used information theoretic insights in driving our deep learning designs. Here, we ask if the deep learning architectures we have learnt can provide an insight to the information theory of communication with feedback. We aim to understand the behavior of Deepcode (i.e., the coded bits generated via RNN in Phase II). We show that in the second phase, (a) the encoder focuses on refining information bits that were corrupted by large noise in the first phase; and (b) the coded bit depends on past as well as current information bits, i.e., coupling in the coding process.

**Correcting highly corrupted noise in Phase I.** The major motivation of the two-phase encoding scheme is that after Phase I, the encoder knows which out of $K$ information bits were corrupted by a large noise, and in Phase II, encoder can focus on refining those bits. In Figure 5, we plot samples of $(n_k, c_{k,1})$ (left) and $(n_k, c_{k,2})$ (right) for $b_k = 1$ and $b_k = 0$ where $n_k$ denotes the noise added to the the transmission of $b_k$ in the first phase. Consider $b_k = 1$. This figure shows that if the noise added to bit $b_k$ in phase 1 is large, encoder generates coded bits close to zero (i.e., does not further refine $b_k$). Otherwise, encoder generates coded bits of large magnitude (i.e., use more power to refine $b_k$).

**Coupling.** A natural question is whether our feedback code is exploiting the memory of RNN and coding information bits jointly. To answer this question, we look at the correlation between

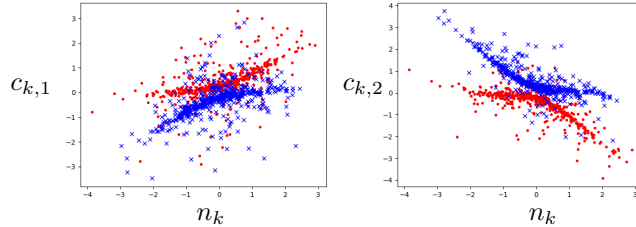

Figure 5: Noise in first phase vs. first parity bit (left) and second parity bit (right). Blue(x) points are when $b_k = 1$ and Red (o) points are for $b_k = 0$.

information bits and the coded bits. If the memory of RNN were not used, we would expect the coded bits $(c_{k,1}, c_{k,2})$ to depend only on $b_k$. We find that $\mathbb{E}[c_{k,1}b_k] = -0.42, \mathbb{E}[c_{k,1}b_{k-1}] = -0.24, \mathbb{E}[c_{k,1}b_{k-2}] = -0.1, \mathbb{E}[c_{k,1}b_{k-3}] = -0.05$, and $\mathbb{E}[c_{k,2}b_k] = 0.57, \mathbb{E}[c_{k,2}b_{k-1}] = -0.11, \mathbb{E}[c_{k,2}b_{k-2}] = -0.05, \mathbb{E}[c_{k,2}b_{k-3}] = -0.02$ (for the encoder for forward SNR 0dB and noiseless feedback). This result implies that the RNN encoder does make use of the memory, of length two to three.

Overall, our analysis suggests that Deepcode exploits memory and selectively enhances bits that were subject to larger noise - properties reminiscent of any good code. We also observe that the relationship between the transmitted bit and previous feedback demonstrates a non-linear relationship as expected. Thus our code has features requisite of a strong feedback code. Furthermore, improvements can be obtained if instead of transmitting two coded symbols per bit during Phase II, an attention-type mechanism can be used to zoom in on bits that were prone to high noise in Phase I. These insights suggest the following generic feedback code: it is a sequential code with long cumulative memory but the importance of a given bit in the memory is *dynamically* weighted based on the feedback.

## 6 Conclusion

In this paper we have shown that appropriately designed and trained RNN codes (encoder and decoder), which we call Deepcode, outperform the state-of-the-art codes by a significant margin on the challenging problem of communicating over AWGN channels with noisy output feedback, both on the theoretical model and with practical considerations taken into account. By concatenating Deepcode with a traditional outer code, the BER curve drops significantly with increasing block lengths, allowing generalizations of the learned neural network architectures. The encoding and decoding capabilities of the RNN architectures suggest that new codes could be found in other open problems in information theory (e.g., network settings), where practical codes are sorely missing.

## 7 Acknowledgment

We thank Shrinivas Kudekar and Saurabh Tavildar for helpful discussions and providing references to the state-of-the-art feedforward codes. We thank Dina Katabi for a detailed discussion that prompted work on system implementation. This work is in part supported by National Science Foundation awards CCF-1553452 and RI-1815535, Army Research Office under grant number W911NF-18-1-0384, and Amazon Catalyst award. Y. Jiang and S. Kannan would also like to acknowledge NSF awards 1651236 and 1703403.

## Footnotes

[1]Source codes are available under `https://github.com/hyejikim1/feedback_code` (Keras) and `https://github.com/yihanjiang/feedback_code` (PyTorch).

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
