[Supplementary Material]

## Appendix

## A  State-of-the art codes used in comparison

In this section, we provide details on how to compute the BER and BLER of state-of-the art feedforward codes. LTE turbo code used in the simulation uses trellis-([13, 15], 13) convolutional code (octal notation) as a component code, and uses quadratic permutation polynomial (QPP) interleaver. Decoding is done by 8 iterations of Belief Propagation (BP) decoder that uses a posteriori probability (APP) decoder as the constituent decoder. Tail-bitting convolutional codes (TBCC) used in the simulation has a constraint length 7 and trellis ([123,135,157]) (in octal notation), and uses Viterbi decoder. Polar code used in the simulation uses success cancellation list decoding (SCL) with list size 8. LDPC code used in the simulation (Rate 1/3, maps 64 bits to a length-196 codeword with sub-matrix dimension 16) uses the parity check matrix shown below, and layered offset min-sum decoder is used with offset parameter 0.22 and (max) iteration 25.

$$
\begin{bmatrix}
10 & 11 & 2 & 3 & 0 & -1 & -1 & -1 & -1 & -1 & -1 & -1 \\
-1 & 15 & 9 & 9 & 14 & 0 & -1 & -1 & -1 & -1 & -1 & -1 \\
6 & -1 & 5 & 13 & -1 & 11 & 0 & -1 & -1 & -1 & -1 & -1 \\
-1 & 5 & -1 & 8 & 12 & -1 & 6 & 0 & -1 & -1 & -1 & -1 \\
-1 & 11 & -1 & -1 & 1 & -1 & -1 & 11 & 0 & -1 & -1 & -1 \\
-1 & 2 & -1 & -1 & 14 & 12 & -1 & 7 & -1 & 0 & -1 & -1 \\
-1 & 15 & 10 & -1 & -1 & -1 & -1 & -1 & 11 & -1 & 0 & -1 \\
-1 & -1 & -1 & 7 & -1 & 11 & -1 & 3 & -1 & -1 & -1 & 0
\end{bmatrix}
$$

## B  Implementation details

In this section, we provide implementation details on the neural encoders and decoders, introduced in Section 3, for the AWGN channels with feedback.

### B.1  Illustration on Scheme A. RNN feedback encoder/decoder (RNN (tanh)).

The details of neural encoder and decoder architectures for RNN feedback code are illustrated in Tables 1 and Figure 6. For both the RNN (tanh) and RNN (linear) feedback codes, we use the same architecture; the only difference is that RNN (tanh) encoder has a tanh activation, and RNN (linear) encoder has a linear activation (for both the recurrent activation and output activation).

*Training.* Adding more details on training, we reduce the learning rate by 10 times after training with $10^6$ examples, starting from 0.02. In measuring the BER, take the average of $10^8$ bits at $-1, 0$dB and $10^9$ bits at $1, 2$dB.

Table 1: Architecture of RNN feedback encoder (left) and decoder (right) for AWGN channels with noisy feedback.

| Layer | Output dimension |
|---|---|
| Input | (K, 4) |
| RNN (linear or tanh) | (K, 50) |
| Dense (sigmoid) | (K, 2) |
| Normalization | (K, 2) |

| Layer | Output dimension |
|---|---|
| Input | (K, 3) |
| bi-GRU | (K, 100) |
| Batch Normalization | (K, 100) |
| bi-GRU | (K, 100) |
| Batch Normalization | (K, 100) |
| Dense (sigmoid) | (K, 1) |

### B.2  Illustration on Scheme B. RNN feedback code with zero padding (RNN (tanh) + ZP).

The encoder and decoder structures with zero padding are shown in Figure 7 and Figure 8, respectively. In training, we use the same method as in training for scheme A. The only difference is that when we evaluate the loss, we evaluate the binary crossentropy loss on the information bits of length $K$ only and ignore the loss on the last padded bit.

### B.3  Illustration on Scheme C. RNN feedback code with power allocation (RNN(tanh) + ZP + W).

The encoder structure for scheme C is shown in Figure 9. The decoder architecture is the same as the decoder for Scheme B in Figure 8. Specifically, we introduce three trainable weights $(w_0, w_1, w_2)$ and let $\mathbb{E}[c_k^2] =$

Figure 6: RNN feedback encoder (left) and decoder (right)

Figure 7: Encoder for scheme B.

Figure 8: Decoder for schemes B,C,D.

$w_0^2$, $\mathbb{E}[c_{k,1}^2] = w_1^2$, $\mathbb{E}[c_{k,2}^2] = w_2^2$ for all $k \in \{1, \cdots, K\}$ where $w_0^2 + w_1^2 + w_2^2 = 3$ (c.f. in Encoder B, we let $\mathbb{E}[c_k^2] = \mathbb{E}[c_{k,1}^2] = \mathbb{E}[c_{k,2}^2] = 1$). In training, we initialize $w_i$s by 1 and train the encoder and decoder jointly as we trained Schemes A and B. The trained weights are $(w_1, w_2, w_3) = (1.13, 0.90, 0.96)$ (trained at -1dB). This implies that the encoder uses the most power in Phase I, to transmit (raw) information bits. In Phase II, the encoder uses more power on the second parity bits than in the first parity bits.

## B.4 Scheme D. RNN feedback code with bit power allocation (RNN(tanh) + ZP + W + A).

The encoder structure for scheme D is shown in Figure 10. The decoder architecture is the same as the decoder for Scheme B in Figure 8. To the full generality, we can train all the weights $\mathbf{a} = a_1, \cdots, a_K$ where $a_k$ is the amplitude of the $k$-th information bit. However, we let $a_5, \cdots, a_{K-4} = 1$ and only train first 4 weights and the last 5 weights, $a_1, a_2, a_3, a_4$ and $a_{K-3}, a_{K-2}, a_{K-1}, a_K, a_{K+1}$. This is because by doing so, we can generalize the encoder to longer block lengths by unrolling and also the BER of bits in the middle have the same error regardless of positions.

Figure 9: Encoder C.

For scheme D, instead of training from the random initialization, we start from the trained model in C and additionally train $\mathbf{a}$ on top of the trained model. (We allow the trained weights change as we learn $\mathbf{a}$. The trained weights are $(a_1, a_2, a_3, a_4) = (0.87, 0.93, 0.96, 0.98)$ and $(a_{K-3}, a_{K-2}, a_{K-1}, a_K, a_{K+1}) = (1.009, 1.013, 1.056, 1.199, 0.935)$ (for $-1dB$ trained model). As we expected, the trained weights in the later bits are larger. Also, the weight at the $K+1$th bit position is small because last bit is always zero and does not convey any information. On the other hand, trained weights in the beginning positions are small because without the power control, these bits were very robust to noise.

Figure 10: Encoder D.

## B.5 Feedback with delay and coding

Practical feedback typically is delayed for a random time, thus the encoder cannot use immediate feedback to encode. The feedback is randomly delayed up to block length $K$, we are restricted not to use feedback till $K$ bits are transmitted. Coding in both forward and feedback channel under noisy feedback will strengthen the reliability of communication.

We propose an active and delayed feedback scheme to overcome noisy feedback and delaying effect, the 1/3 code rate encoder is shown in Figure 11. In the first phase, the $K$ information bits can be encoded by Bi-GRU, while the feedback is delayed and can only be used in the next phase. The second and third phase uses uni-directional GRU to encode with $K$-delayed feedback, which means at index $m$ of phase 2, the encoder can only use the feedback before of index $m$ of phase 1. Receiver side encode the feedback by unidirectional GRU and send through the delayed feedback channel back to the transmitter. The decoder is a Bi-GRU which waits to decode until all information bits are received.

We can see from Figure 12 (Right), passive feedback under delayed feedback still has better performance compared to the turbo code, and beats S-K code under high SNR regimes. The delaying effect is enabled via our RNN feedback coding scheme. The gain is from: (1) adding an additional phase, which gives the RNN more fault tolerance comparing to 2-phase coding; (2) training the RNN to decode with delayed feedback.

Figure 12 (Left) shows the performance under noisy feedback. The forward channel is under AWGN 0dB, while the x-axis shows the feedback SNR. The C-L and S-K code fail to decode under noisy feedback channel. Passive feedback code achieve better performance comparing to C-L and S-K code, while active feedback code outperform passive feedback code. The performance gain is from: (1) the coding gain of active feedback, which gives the encoder RNN better robust representation of feedback code; (2) as the feedback is noisy, delayed coding actually averages the noise, which leads to better performance.

Figure 11: Encoder for delayed feedback

Figure 12: Neural schemes for delay with feedback under noisy (left) and noiseless (right) feedback

**Literature on coded feedback** In [37], the authors show that active feedback can improve the reliability under noisy feedback if the feedback SNR is sufficiently larger than the forward SNR. Their coding scheme assumes that the encoder and decoder share a common random i.i.d. sequence (of length equals to the coded block length), mutually independent of the noise sequences and the message, which we do not have, which makes it hard to compare our scheme with theirs.

# C   Concatenation of Deepcode with existing codes

Concatenated codes are constructed from two or more codes, originally proposed by Forney [38]. We concatenate forward error correcting codes (that do not use a feedback) with our neural code that makes use of feedback. Encoding is performed in two steps; we first map information bits into a turbo code, and then encode the turbo code via an encoder for channels with feedback. Decoding is also performed in two steps. In the first step, the decoder recovers the estimates of turbo codes. In the second step, the decoder recovers information bits based on the estimates of turbo codes. For the experiment in Section 4, for which results are shown in Figure 4 (Right), we use the rate 1/3 LTE turbo code as an outer code; LTE turbo code uses ([13, 15], 13) convolutional code (octal notation) as a component code. We compare the performance of the concatenated code with a rate 1/9 turbo code, which uses ([13,17,16,15,11],13) convolutional code as a component code (introduced in [39]). Besides turbo codes, any existing codes (e.g., LDPC, polar, convolutional codes) can be used as an outer code. We also note that C-L scheme is based on the concatenation idea [7].

# D   Existing codes: C-L and S-K schemes

In this section, we provide an illustration of two baseline schemes, C-L scheme and S-K scheme, and the connection between these schemes and our neural codes.

A simple scheme is to linearly encode each information bit separately using feedback. For each bit $b_k$, the encoder generates three coded bits $(c_{k1}, c_{k2}, c_{k3})$. This is the Chance-Love scheme proposed in [7]. One of the contributions of [7] is to empirically find the optimal weights for the linear functions (there is no closed-form solution). Another contribution is that they propose concatenating their code with an existing forward error correction code such as turbo codes, i.e., instead of mapping the information bits **b** directly to the codeword **c**, the encoder maps **b** to a turbo code **d** and then map the turbo code **d** to a codeword **c**.

Can we start with a neural architecture that includes the C-L as a special case and improve upon it? Due to the sequential nature of feedback encoder, recurrent neural network (RNN) architectures are natural candidates. A simple neural architecture that includes the C-L scheme as a special case is illustrated in Figure 14. We consider various versions of RNN encoders –RNN with linear activation functions, and nonlinear RNN, GRU, LSTM. We train the encoder and decoder jointly. For all architectures, we use 50 hidden units. The BER of trained networks are shown in Table 15. We can see that the BER of nonlinear RNN is smaller than a linear feedback scheme with weights optimized.

Figure 13: Illustration of encoding of $k$-th bit for a rate-1/3 linear encoder in Chance-Love scheme

Figure 14: Encoding of $k$-th bit for a rate-1/3 RNN encoder

| Scheme | BER at $1dB$ ($\sigma_F^2 = 0.01$) |
|---|---|
| Bit-by-bit linear (Shalkwijk-Kailath) | 0.0023 |
| Bit-by-bit linear (Chance-Love) | 7.83e-04 |
| Bit-by-bit linear RNN | 0.0046 |
| Bit-by-bit RNN | **1.56e-04** |
| Bit-by-bit GRU | 1.58e-04 |
| Bit-by-bit LSTM | 1.88e-04 |

Figure 15: BER of other RNN architectures. Rate 1/3

Although RNN has the capability to represent any linear bit-by-bit linear encoder/decoder, we can see that the training is highly nontrivial, and for linear RNN, the neural network converges to a local optima. On the other hand, for nonlinear RNNs, the trained encoder performs better than the weight-optimized linear scheme.

From coding theory, we know that the bit error rate should go down as block length gets longer. If we use bit-by-bit encoding, the improvement can never be realized because BER remains the same now matter how long the block is. In order to enable the bit error to decay faster as block length increases, the encoder has to code information bits *jointly*. A celebrated feedback coding scheme, Shalkwijk–Kailath scheme, simplified/illustrated in Figure 16, belongs to this category.

Figure 16: Illustration of S-K encoder

**S-K scheme.** Here all information bits are used only to generate the first codeword. The rest of the codewords depend only on the feedback (noise added to the previous transmission). Although S-K scheme does encode all information bits jointly, transmitting all information bits in the first phase requires a high numerical precision as block length increases. For example, for 50 information bits, the transmitter transmits $\sum_{k=1}^{K} b_k 2^k$ (with a power normalization and subtracting a constant to set mean to be 0).

Our approach is different from S-K scheme in that we aim to use the memory of RNN to design an encoder that encodes the information bits jointly. Since RNN has a memory in it, naturally it allows encoding bits jointly. The challenge is whether we can we find/train a neural network encoder which makes use of the RNN memory. For example, Figure 17 illustrates a somewhat natural architecture we attempted. However, after training, the BER performance is only as good as the BER of bit-by-bit encoding, which means that the memory in the RNN is not being successfully used to jointly encode the information bits.

Figure 17: Bit-coupled RNN encoder

# E Robustness under bursty Gaussian channels

Bursty Gaussian channel is a channel where there is a background Gaussian noise $n_{1i}$, and occasionally, with a small probability $\alpha$, a Gaussian noise with high power (bursty noise, $n_{2i}$) is added on top of the background noise. Mathematically, we consider the following bursty Gaussian channel: $y_i = x_i + n_i$, where

$$n_i = n_{1i} + e_i n_{2i},$$
$$n_{1i} \sim \mathcal{N}(0, \sigma_o^2), \ \ n_{2i} \sim \mathcal{N}(0, \sigma_1^2), \ \ e_i \sim Bern(\alpha).$$

We test the robustness of our feedback code under bursty Gaussian channel. Figure 18 shows the BER as a function of $\alpha$ (probability of having a burst noise), for $-1dB$, and two different power of burst noise. We choose $\sigma_0^2$ so that $\alpha\sigma_1^2 + \sigma_0^2 = \sigma^2$ (i.e., we keep the total power of the noise). As we can see from the Figure, as $\alpha$ increases, the BER decreases, showing that under bursty noise, the bit error rate is smaller.

Figure 18: Deepcode is robust to bursty noise.