[Reviews · NeurIPS 2018]

Reviewer 1



I acknowledge that I read the author's response, which address many of my concerns. As a result, I am raising my score to a 5 -- I lean toward rejection but not strongly. I remain concerned about the absence of a related work section -- and the related work cited in the response still seems inadequate. In our discussions, R2 pointed out a lot of additional work the authors appear to have missed. ----- Summary: This paper describes a deep learning approach to encoding and decoding in a noisy channel with feedback, something that has been little explored in the literature to date. The formal noisy channel setting is similar to a standard autoencoder framework, with a few key differences. For one, we usually encode and transmit one bit of a message at time due to channel limits, and second we get feedback, usually in the form of a noisy version of each encoded bit. Due to the sequential nature of the problem, plus the availability of feedback, the authors apply an RNN architecture. The input to the decoder at each step is the next bit to encode plus an estimate of the noise from previous steps (derived from the difference between the encoded message and the received feedback). Experiments suggest that this approach significantly outperforms existing approaches. The authors explore several variants of the RNN architecture specific to the setting: normalization to restrict the power of the encoded message, per-bit weights to adapt the power of each encoded message (similar to an attention mechanism), and an encoder-decoder framework for the feedback. On the one hand, I think this paper is pretty solid and could be accepted on the merits of its work. On the other hand, the paper lacks entirely a related work section and fails to discuss ANY previous work on deep learning applied to noisy channels. A quick Google search indicates there is plenty of such work (as does the mysterious presence of uncited papers in the references, see below). It is hard to imagine accepting the paper in its current form. I would like to hear the authors' explanation and discuss with the other reviewers before making a final decision. For now I will put a temporary "clear reject" score but am willing to revise it. Quality: Although a straightforward application for the most part, the work nonetheless appears high quality to me. The choice to apply an RNN seems sound and the various proposed architectures seem well-matched to the problem. The authors demonstrate a thorough understanding of the problem setting (though their description could be improved, see below). The results appear to speak for themselves: this approach soundly beats the baselines. I especially appreciate Sections 4 and 5 -- the experiments in this paper are thorough, and the authors make a good faith effort to provide insights about their proposed model's behavior, rather than just reporting quantitative performance. A few questions about the proposed approach and experiments: - The authors need to justify the choice to use an RNN without first experimenting with a non-recurrent architecture. Do we really need to capture longterm dependencies? The coupling results (lines 313-318) only cover a very short history: we could easily design an MLP or convolutional architecture that accepts multiple steps of input to capture such history. Did the authors try this? - The discussion of complexity (lines 234-243) is somewhat vague. I expected to see an actual complexity analysis (in terms of block length and maybe coded message length?) and then maybe some empirical timing experiments -- along with a comparison to existing algorithms (in both formal analysis and in the experiments). Instead we got a vague handwave suggesting that they're comparable. That's not very convincing. Clarity: This paper is for the most part well-organized and written (modulo the missing related work), but it may be difficult for readers who are unfamiliar with coding theory. To make it more accessible to a wider reader audience (thus increasing its potential impact), I suggest the following: - clearly articulate how this problem differs from, say, standard autoencoding problems or machine translation, which both have similar looking structure. - show the full end-to-end architecture with encoder AND decoder and explain how it works. It's still not clear to me whether the decoder is run once at the end, after the full coded message is transmitted (as implied by the bidirectional RNN architecture) or whether each decoding step happens after the corresponding encoded step is transmitted. Some nitpicks: - The justification for the first phase is hard to follow. Why do we want to send the un-encoded bits first? What would happen if we omitted that phase (maybe some results for an RNN with phase 1)? - Why does the RNN in Figure 2 output two coded bits per step? Is that simply due to the rate of 1/3 per the equation in lines 164-165 -- so that with a different rate, we'd output more codes? - Why is the input to the RNN at each step the difference between the feedback and codes, rather than just the raw feedback? Did the authors try this? - What is meant by "unrolling" the RNN cells when working with longer block lengths (line 284)? Does this mean using truncated backprop through time or what? - The description of the architecture used (lines 239-241) is a bit confusing: the authors say they used an RNN in the encoder but a GRU in the decoder. Is the encoder in fact a vanilla RNN or is this a typo? Originality: It is impossible to judge the paper's originality in the absence of a thorough discussion of related work. Why is there no related work section, or at least a discussion of relevant previous work in the introduction? This is especially baffling given that the references section lists multiple relevant papers [10-17] that are not cited anywhere in the paper. This means they're listed in the compiled bibtex database file, which in turn means that at some point, a version of the paper citing them must have been compiled. Was the related work section accidentally commented out? On the other hand, the paper is exactly 8 pages long, meaning that there's no room for related work anyway. In any case, the paper's language, the lack of discussion of similar work, and the absence of other neural net baselines, implies this work may be the first to apply RNNs to the noisy channel with feedback problem. I am not sufficiently familiar with work in this area to confirm or deny that claim, but I am dubious of any claim to be the first to apply deep learning to a classic problem -- and the works listed in the references in this paper [10-17] (plus those turned up in a quick search) cast some doubt. The authors need to tighten their claim: what exactly is their original contribution? And they need to position their work with respect to related research. If there are alternative approaches to this problem -- either previously published or even not published but obvious -- then they need to make a best effort to compare them rhetorically and empirically. Significance: The significance of an applied paper of this type is tied almost entirely to its empirical results and what new insights we can derive from them. Taking the paper at face value, it appears quite significant -- the proposed RNN architecture is substantially better than standard approaches, and I -- for one -- did not know that, even if it seems unsurprising. However, given the omission of a related work discussion, it is impossible to fully decide the significance of this paper. If someone else previously applied RNNs to this problem and produced similar results, then this paper's contribution is diminished.

Reviewer 2



This submission presents an RNN based construction for channel codes with feedback, a classic code-design problem from information theory. The goal is to design error correcting codes with feedback for the AWGN channel where each transmitted real value is received with added Gaussian noise. In feedback codes, we assume that before transmitting each symbol the transmitter has access to all noisy symbols seen by the receiver, and it can use this information to design the next symbol. Furthermore, codes for an AWGN need to be designed so that the average power (two-norm square/length) of each codeword must be below a threshold. This is a well-studied problem in information theory along with the seminal Schalkwijk-Kailath (S-K) known to be asymptotically rate optimal. However, at fixed blocklengths, the performance of S-K scheme seems quite suboptimal. Moreover, S-K scheme falls short of acceptable performance when the feedback is noisy or delayed. The current submission presents a code design scheme based on RNNs that shows promise in simulations to overcome these shortcomings. The proposed construction is developed in steps, leading to the final construction. We summarise the features of the final construction below: 1. Uncoded bits b_1, ..., b_K are sent (using real values +-A of fixed amplitude A) followed by 2K transmissions with the pair (c_{k,1}, c_{k,2}) chosen to depend on (i) input c_k corresponding to b_k; (ii) the estimated noise y_k - c_k for this transmission; (iii) noise estimates of the previous two transmissions in the second phase y_{k-1,1}-c_{k-1,1} and y_{k-1,2}-c_{k-1,2}. Specifically, a single directional RNN is chosen with the k-th RNN cell generating the input (c_{k,1}, c_{k,2}) based on the observations mentioned above. For the case when the feedback itself is noisy, y_k is replaced by its noisy version \tilde{y}_{k}. Finally, the average power constraint is maintained by normalising the RNN outputs (this is not very clear to me -- do the constraints need to be maintained on average or in the worst-case?) 2. For decoding, the k-th input bit is decoded using the three receptions (y_k, y_{k,1}, y_{k,2}) containing information about it. This decoder comprises a Gated Recurrent Unit (GRU) which is training jointly with the RNN encoder of the previous step, with cross-entropy as the loss function. 3. The first enhancement done to the architecture above is including a zero padding after the message to be transmitted. This is a standard technique in sequential code design that allows one to reduce the error in the final message bits. 4. Next, to combat the unequal error protection offered to the uncoded transmission c_k and the coded transmission (c_{k,1}, c_{k,2}) , the inputs for these information symbols are weighed differently with weights trained empirically. 5. Finally, even the weights (input amplitudes) assigned across different information symbols are set to be different and the weight vector is trained empirically. This final construction is termed a "neural code construction" and numerical evidence is provided to show that this outperforms the S-K scheme in error performance by orders of magnitude at blocklength ~150 (corresponding to sending 50 message bits). Furthermore, the gains are enhanced even more when practical variants such as noisy and delayed feedback are included. The numerical results are interesting and add to the belief that neural network based code designs can compete with the classic code designs, which has been pushed by a recent line of work (see Kim et. al. "COMMUNICATION ALGORITHMS VIA DEEP LEARNING" at ICLR 2018). While the results are interesting and should attract attention from communication industry, who will be happy to focus on a single architectural design that can capture everything from communication to intelligence, I find the contribution incremental. The machine learning tools of RNN used are standard and have already been explored in the past (see for example the ICLR work mentioned above). Further explorations along this theme must be reported in the communication and code design literature where the audience can weigh-in on the nuances arising from considering different use-cases of the same basic idea. Also, from an information-theory point of view, the results presented are unsatisfactory for the following two reasons: 1. Since the paper is presenting results for finite blocklength, they should benchmark their performance against the finite blocklength rate bounds for feedback codes which are available in the literature. S-K code is an asymptotically optimal scheme presented 50 years ago, there is no point in benchmarking against it at lengths as short as 150. 2. The paper extends their proposed codes to longer lengths simply by concatenating it with the optimal S-K scheme. This offers nothing more than the classic S-K scheme; I suspect that many simple list decoding schemes when concatenated with S-K will yield the same performance. Overall, I recommend rejection since there is already a lot of published work on this topic over the past one year and the current submission offers nothing much over those works, and also, since the performance has not been compared with the information theoretically optimal benchmark available in the literature. On the other hand, I believe that studies such as this will be apt for publication in communication conferences where this theme can now be picked up. ==== I thank the authors for their response. Other reviewers have convinced me that the observed performance of RNN based feedback codes proposed in this paper is promising and may lead to further advances in this direction. I have upgraded my rating to 6.

Reviewer 3



There seems to be an increasing amount of interests in improving the design of communication systems using neural networks. Most of the papers in this direction, at least to the knowledge of the current reviewer, are rather simplistic (if not like laughing stocks) to serious communication engineers. The current paper is the first exception the current reviewer has witnessed. The paper proposes an RNN architecture to learn a feedback code over the additive white Gaussian noise channel with noisy feedback. Using the proposed approach, the paper constructs a concrete rate 1/3 (150, 50) feedback code that seems to perform better than a few existing codes. The paper also exhibits practical considerations such as delayed (noisy) feedback and longer block lengths. Although the original architecture is rather simple, how these additional components can be added is carefully studied. There are a few reasons why the paper, despite the highest praise in the opening sentence of this review, did not receive a higher score. First, a more recent result [27] was not compared to the current development (although [27] is mysteriously in the reference list). Instead, the paper keeps going back to Schalkwijk-Kailath, which was proven to fail miserably [8]. Overall, the paper is somewhat backward-looking in the literature of feedback communication. There are many recent results discussing better coding schemes for noisy feedback in the literature. Even if the results are comparable to the state of the art, the approach itself is quite interesting, so the paper should be less defensive and strive to include the best recent results. Second, the focus is clearly on the communication theory side, not on the network development side. For that reason, the current reviewer is ambivalent on whether this contribution fits a communication theory venue better. It's a tough balancing job to adequately explain both of the network architecture and the problem it solves. The network architecture in the current paper is a bit too thin, so as a NIPS paper, the current reviewer would like to see more on that side, not on the nitty gritty communication engineering issues. I would be a good idea to decide if the paper is for NIPS or for ISIT/GLOBECOM. If the former, which is apparently the case, it should be written in a way that is more appealing to the NIPS audience.